# Cross-Linguistic Failures and Disparities in Western LLM Medical Reasoning: Analyzing XMedBench and CrossMMLU Across Western and Non-Western Languages

**Rehan Nazeem, Akira Hoque, Vedesh Ray Peddoddi[†], Tim Liu[†], Kevin Zhu** [*]

## Abstract

As Western Large Language Models (LLMs) are increasingly considered for clinical decision support, their cross-lingual reliability remains uncertain. Strong Western performance is often assumed to generalize, yet has rarely been tested in non-Western languages where medical AI may alleviate ongoing healthcare inequalities. To investigate disparities, we evaluated GPT-4o, two Anthropic models (Claude 3.5 Sonnet and Claude 4.5 Sonnet), and two Gemini models (Gemini 1.5 Pro and Gemini 2.0 Flash) on two multilingual benchmarks: XMedBench, assessing clinical reasoning in six languages, and CrossMMLU, testing a variety of knowledge across seven languages. All models were run with a unified multiple-choice prompting technique and deterministic decoding to enable controlled comparisons. We find substantial cross-language variability. Several mid- and low-resource languages showed declines, especially on questions requiring spatial anatomical understanding or culturally specific terminology. English did not consistently yield the highest performance. This variance appears dataset-dependent, as observed in XMedBench results, suggesting that disparities may arise from model training biases and domain limitations. Across datasets, consistent weaknesses emerged in embryology-related questions. Overall, our findings show that multilingual clinical question answering remains uneven in primarily western developed LLMs, emphasizing the need for linguistically inclusive evaluation and mitigation strategies.

## 1 Introduction

Large Language Models have rapidly advanced over the last decade and are tools capable of assisting in healthcare efficiency Maity & Saikia (2025). Their growing presence in healthcare has sparked hopes for expanding medical guidance to regions where trained doctors are scarce Maity & Saikia (2025). However, this remains doubtful, as most LLMs are evaluated primarily in English and other Western languages, creating a blind spot in how well they function for billions of speakers of non-Western and low-resource languages Beyene et al. (2025). To investigate this gap, we analyze two complementary datasets: XMedBench, which evaluates medical reasoning across a variety of languages, and CrossMMLU, which extends general scientific and humanities knowledge into a multilingual setting to compare medical-related performance to general LLM performance Mahajan et al. (2025) Wang et al. (2024). This enables us to determine whether current models behave consistently across linguistic boundaries and medical content domains.

## 2 Related Work

LLM multilingual question-answering has become a key area of study, used to assess cross-language accuracy and performance disparities. Past benchmarks such as MGSM Shi et al. (2022) and TyDi QA Clark et al. (2020) demonstrate that Western languages consistently achieve higher average

---

[*][†]These authors contributed equally as third authors.

performance than non-Western or low-resource languages. Such errors may result in fatal consequences in medical contexts Bélisle-Pipon (2024). To identify these gaps, prior work introduced benchmarks like MMLU, proposed by Hendrycks et al. to evaluate LLM accuracy across academic and professional domains. XMedBench, published in 2024 by Wang et al. in Apollo: A Lightweight Multilingual Medical LLM towards Democratizing Medical AI to 6B People, was introduced as a low resource language clinical benchmark. The dataset includes six widely to semi-widely spoken languages: English, Spanish, Chinese, Arabic, Hindi, and French. The work discusses the limitations of English-dominated medical knowledge and how these failures disproportionately impact underserved communities. Further work, such as Improving the Multilingual Fairness of Language Models with Adaptive Gradient-Based Tokenization Ahia et al. (2024), explores how tokenization bias disadvantages low-resource languages. Over-segmentation of non-Latin scripts leads to increased computational costs and fragmented semantic understanding, reinforcing English as the primary language for LLM interaction.

## 3 METHODS

For medical-domain analysis, we used the XMedBench benchmark, which consists of multiple-choice medical examination questions in six languages as discussed previously. The dataset contains 7,327 English questions; 8,759 Chinese questions (8,018 tested); 321 French questions (278 tested); 2,742 Spanish questions; 1,088 Arabic questions; and 1,089 Hindi questions. Some questions were excluded due to filtering constraints, and only questions with four answer choices were evaluated. We evaluated some of the most popular western-developed LLM's: GPT-4o, Claude 4.5 Sonnet, and Gemini 1.5 Pro using API access with a temperature of 0. During experimentation, Gemini 1.5 Pro was deprecated and replaced with Gemini 2.5 Pro due to architectural similarity. For each language, we processed up to the first 300 questions, presenting four answer options labeled A–D and concluding with the instruction: "Please respond with only the letter of the correct answer (A, B, C, or D)." Model outputs were compared against gold answers, and incorrect responses were manually reviewed. To assess general knowledge across languages, we used the CrossMMLU benchmark, which extends MMLU into a multilingual setting across STEM and humanities subjects. We evaluated Anatomy, Clinical Knowledge, Professional Medicine, Medical Genetics, Nutrition, College Biology, College Chemistry, Sociology, High School Psychology, Philosophy, Moral Scenarios, World Religions, Geography, and Global Facts. Each language was tested on 640 questions (40 per subject) across seven languages: English, Chinese, Spanish, Malay, Indonesian, Vietnamese, and Filipino. All models were evaluated using deterministic decoding. Each language was evaluated on a capped sample size to allow for unbiased comparisons across models and increase statistical power. Furthermore, languages like French contained fewer questions in total, limiting our available sample count. Therefore, our sample size may introduce variance given dataset limitations from XMedBench and MMLU.

## 4 DATASET OVERLAP ANALYSIS

Both XMedBench and CrossMMLU contain overlapping domains including clinical physiology, anatomy, microbiology, pharmacology, and procedural reasoning. Clinical physiology questions often require multi-step reasoning involving unit conversions, biological systems, and numerical estimation. Both datasets assess procedural reasoning related to infection control, sterilization, and patient handling protocols. Pharmacological computation tasks involving dosage conversion, infusion rates, and medication timing appear in both benchmarks. Anatomical questions test spatial and functional relationships such as joint movement classification and landmark identification, challenging unimodal models to perform implicit spatial reasoning. Microbiology questions assess organism classification and infection identification, while patient safety questions evaluate clinical judgment in practical scenarios.

## 5 RESULTS

When comparing Western and non-Western languages, we observed unexpectedly lower accuracy in several Western languages on XMedBench. English exhibited accuracy between 80–85 percent, while Spanish consistently achieved approximately 90 percent. This may be due to the questions

being derived from country-specific medical examinations rather than translations, reflecting how conventions and terminology usage across the datasets may have impacted model accuracy. Under Claude 3.5 Sonnet, French exhibited the weakest performance, suggesting instability due to small sample size. Hindi achieved accuracy comparable to English despite expectations of lower performance, while Arabic showed the lowest average accuracy. In contrast, CrossMMLU demonstrated relatively uniform performance across languages, with English achieving the highest accuracy (83–84 percent) followed closely by Spanish (81–82 percent). These differences were marginal, suggesting that medically specific questions in XMedBench amplify language-dependent failures. Performance also varied by domain and question type. Medical domains such as craniofacial identification, anatomical terminology, and clinical procedures consistently exhibited lower accuracy. Subject-level analysis revealed that GPT-4o performed best on kidney anatomy and molecular genetics, Gemini excelled in calculation-based questions, and Claude 3.5 Sonnet performed best on ecological and mandibular reasoning. Across both datasets, embryology consistently showed the highest failure rates, likely due to spatial and visual reasoning requirements beyond text-based modeling.

## 6 CASE STUDIES

In a dosage-conversion question requiring multiplication followed by unit conversion, GPT-4o answered correctly while Claude and Gemini produced answers with incorrect unit conversion. This suggests that while all models performed the initial multiplication, only GPT-4o reliably handled unit conversion. In a metabolic energy question, Gemini incorrectly assumed an oxygen-to-energy conversion factor, resulting in a fivefold underestimation. This failure reflects a lack of physiological knowledge rather than numerical reasoning limitations.

## 7 DISCUSSION

A major challenge in low-resource languages is insufficient high-quality domain-specific training data. One potential solution is the SynDARin pipeline Ghazaryan et al. (2025), which generates synthetic QA pairs by translating high-quality English medical questions into target languages. Retrieval-Augmented Generation (RAG) is another promising approach, grounding responses in external resources at inference time. While RAG reduces hallucinations, it requires computational infrastructure often unavailable in underserved regions. Lightweight systems may address this. Architectural solutions include Mixture-of-Experts (MoE) models and adapter-based fine-tuning. MoE models activate specialized subnetworks based on input, while adapters enable efficient domain specialization without retraining entire models.

Our study is limited in several ways that restrict the generalization of our findings. We evaluated two datasets (XMedBench and Cross-MMLU), which, despite being complementary, do not fully assess LLMs' linguistic diversity across all possible LRLs. Also, we were not able to use the same Gemini model in both as Gemini-1.5-Pro was deprecated during our research process. This was the same in regards to the same Anthropic Model. Language coverage did not fully align across datasets, perhaps not fully catching some nuances between LLM's knowledge of various LRLs like we anticipated. Domains like embryology and spatial anatomy are also both naturally difficult for LLMs that can only think in text, implying their faults may be attributed to modality limitations rather than just language abilities. Future work should consider a broader range of LLMs by development cost and region to provide more comprehensive results.

## 8 CONCLUSION

We presented a cross-linguistic evaluation of LLMs across medical and general knowledge benchmarks. Despite strong overall reasoning capabilities, models consistently failed in embryology, drug dosage, and procedural reasoning across languages. Errors stemmed from linguistic ambiguity, domain complexity, and architectural limitations rather than language alone. We propose mitigation strategies including synthetic data generation, improved RAG frameworks, and MoE architectures. Future work should expand language coverage, refine evaluation metrics, and construct culturally grounded datasets with clinician input to ensure safety and equity in global medical AI.

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

## A  APPENDIX

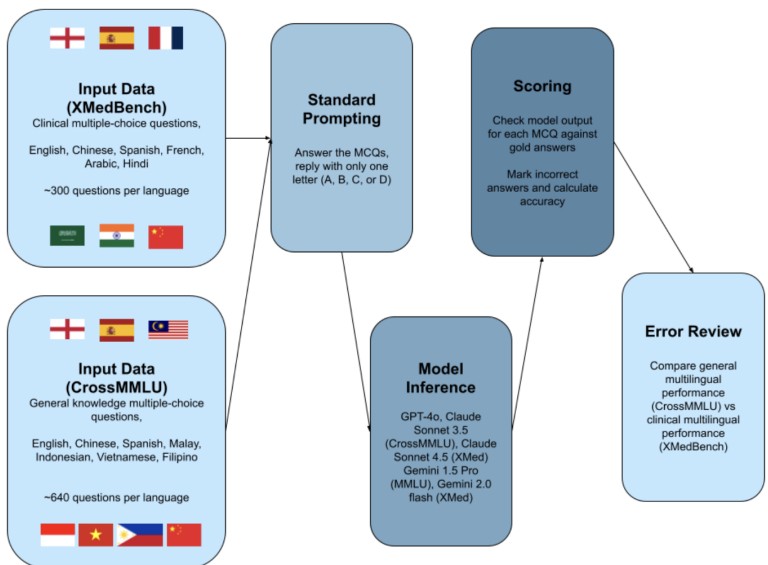

Figure 1: Experimental Pipeline Diagram

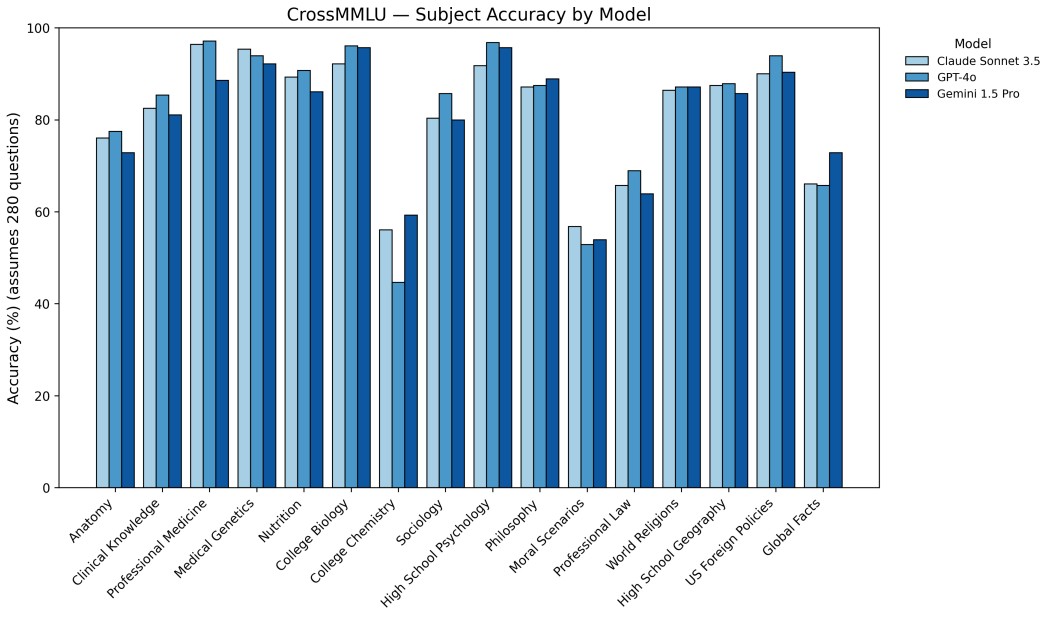

Figure 2: CrossMMLU Accuracy by Subject

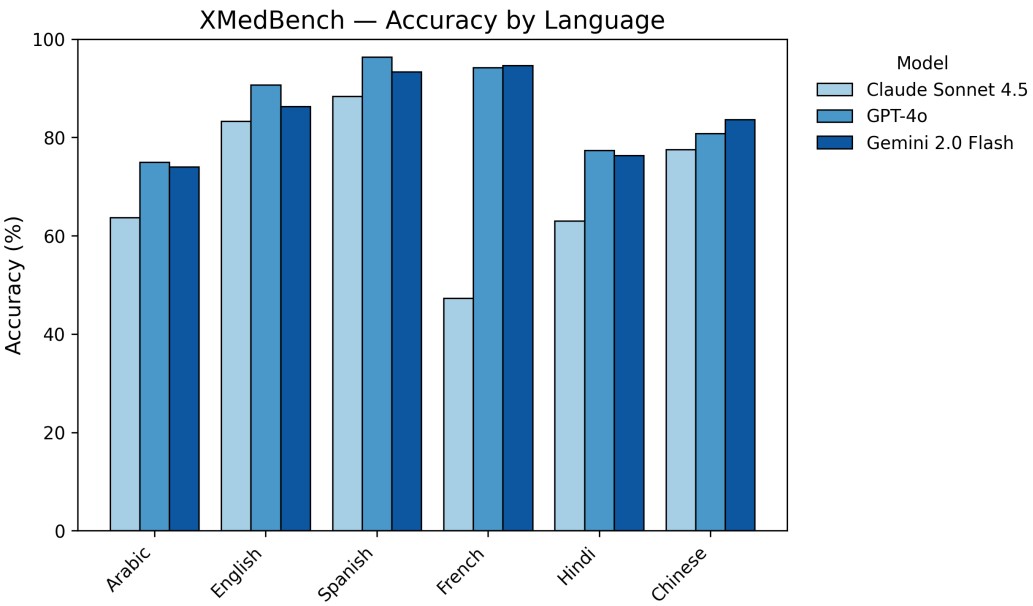

Figure 3: XMedBench Accuracy by Language

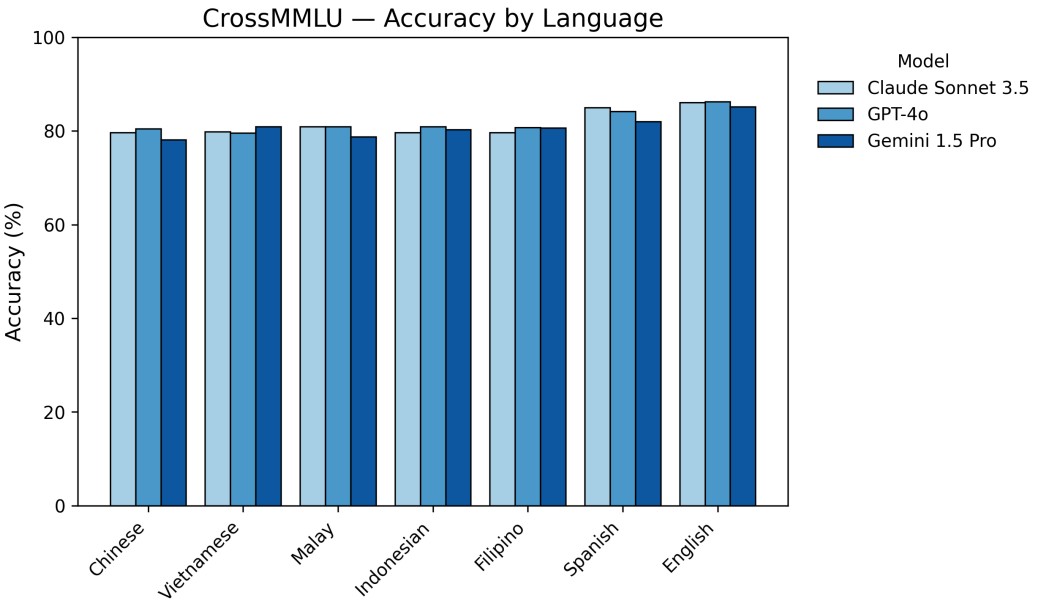

Figure 4: CrossMMLU Accuracy by Language

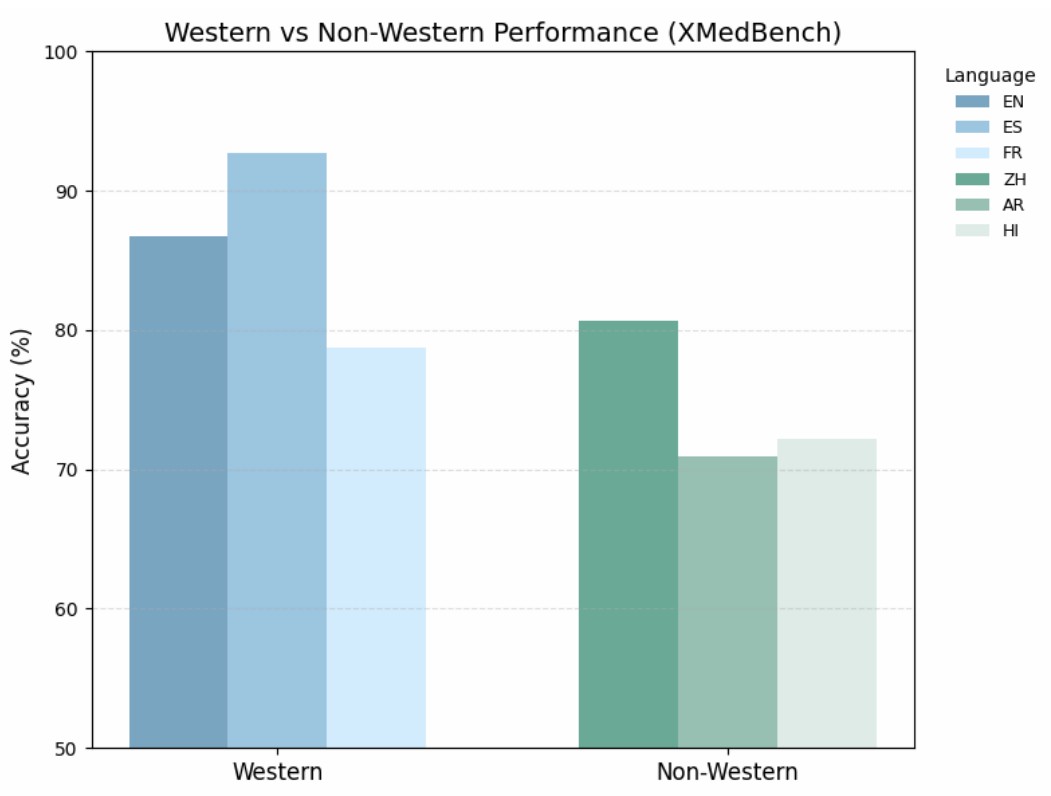

Figure 5: Western Vs Non-Western Performance on XMedBench

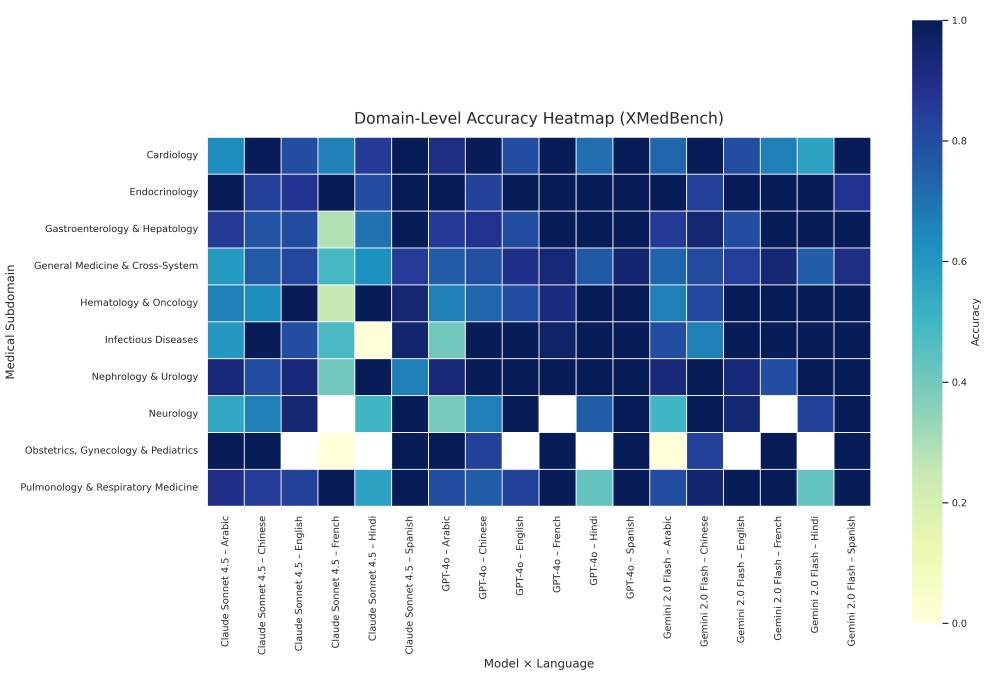

Figure 6: XMedBench Accuracy by Domain Heatmap

Dosage Calculation

Model Responses

Numerical Dosage Question

A patient weighs 62 kg and is prescribed 15 mg/kg of a medication.

How many grams is the required dose?

Correct Answer: 0.93 g

| Model | Response | Evaluation |
|---|---|---|
| GPT-4o | 0.93 g | Correct |
| Claude Sonnet 3.5 | 930 mg | Incorrect |
| Gemini 1.5 Pro | 9300 mg | Incorrect |

*Claude: Correct multiplication, failed unit conversion
*Gemini: Magnitude + unit conversion error

Figure 7: Numerical Dosage Question

Clinical Protocol

Model Responses

Clinical Protocol Question

How often should an intravenous cannula be flushed according to standard clinical protocol?

Correct Answer: every 8 hours

| Model | Response | Evaluation |
|---|---|---|
| GPT-4o | every 8 hours | Correct |
| Claude Sonnet 3.5 | every 8 hours | Correct |
| Gemini 1.5 Pro | every 24 hours | Incorrect |

*Gemini: Misinformed response

Figure 8: Clinical Protocol Question

