# OpenReview forum: "Cross-Linguistic Failures and Disparities in LLM Medical Reasoning: Analyzing XMedBench and CrossMMLU Across Western and Non-Western Languages"
_ICLR.cc/2026/Workshop/AFAA — AFAA 2026 Poster_

### Official Review · Reviewer_t6of · 2026-02-21
**The selection of models need to be as diverse as the languages to arrive at the present conclusion**

**Rating:** 3
**Confidence:** 4

**Summary:**

This paper investigates the cross-lingual reliability of large language models in clinical decision support contexts.

**Strengths:**

The authors evaluate five models: GPT-4o, Claude 3.5 Sonnet, Claude 4.5 Sonnet, Gemini 1.5 Pro, and Gemini 2.0 Flash across two multilingual benchmarks: (1) XMedBench, which covers clinical reasoning in six languages, and (2) CrossMMLU, which spans general and medical knowledge domains across seven languages. Using a standardized multiple-choice prompting strategy with deterministic decoding, the study finds substantial performance variability across languages, with mid- and low-resource languages showing notable declines, particularly in spatial anatomical reasoning and culturally specific terminology. Interestingly, English does not consistently yield the highest performance. The authors also identify consistent cross-model weaknesses in embryology-related questions and suggest that observed disparities likely stem from training data biases rather than fundamental model limitations.

**Weaknesses:**

While the authors deserve credit for selecting benchmarks that span a genuinely diverse set of languages, including Filipino, Vietnamese, Malay, and Hindi, the model selection does not reflect this same commitment to diversity. All models evaluated are developed by Western organizations whose training corpora are well-documented to be heavily English-centric. Notably absent are models such as Qwen or DeepSeek, which would provide a more competitive and linguistically appropriate baseline for Chinese; Sarvam or other Indic-focused models for Hindi; or any SEA-region model for Filipino, Malay, and Vietnamese.
This mismatch between dataset diversity and model selection is a substantive methodological concern. The findings, as presented, risk being interpreted as a general characterization of LLM multilingual capability, when they are more accurately a characterization of how Western-developed LLMs handle non-Western languages. Any conclusions about model performance in these languages should be interpreted with this caveat in mind. The authors are encouraged to either include regionally appropriate models in the evaluation or to explicitly reframe their contribution as an audit of Western LLMs on multilingual benchmarks, which is itself a valid and useful contribution, but requires clearer framing.

---

### Official Review · Reviewer_c3f7 · 2026-02-21
**A timely and critical evaluation in AI healthcare and accessibility space**

**Rating:** 3
**Confidence:** 4

**Summary:**

This paper presents a cross linguistic evaluation of LLMs on multilingual benchmarks Xmedbench and CrossMMLU. It captures variability in performance across languages. They process only first 300 questions on GPT 4o , claude 4.5 sonnet, and gemini 2.5pro for each language. they present that multilingual medical QA remiains uneven.

**Strengths:**

- Important and timely topic, AI in healthcare is a crucial field, multi-linguistic healthcare is critical for the world.
- Dataset overlap analysis between XMedBench and CrossMMLU is good, but you can use it further to derive more statistics on how each domian varies if so by language.

**Weaknesses:**

- only the first 300 questions for each language is insufficient especially as some languages have few questions to begin with.
- Statistics and confidence intervals not shown
- Dosage calculation, the prompt does not seem to have units specified, but a valid answer has been marked incorrect. ex 930mg and 0.93g.

---

### Official Review · Reviewer_V3z6 · 2026-02-22
**An important and fairness-critical evaluation study that highlights real multilingual disparities in medical AI, but methodological tightening and statistical rigor are needed to substantiate causal conclusions and strengthen policy relevance.**

**Rating:** 4
**Confidence:** 3

**Summary:**

This paper investigates cross-linguistic disparities in LLM medical reasoning using XMedBench and CrossMMLU, showing that performance varies substantially across languages and that medically specific, spatially grounded domains may amplify failures. The findings highlight that strong English performance does not reliably transfer to non-English medical contexts. The work raises important fairness and deployment concerns for multilingual clinical AI systems.

**Strengths:**

- Directly addresses equity and fairness in high-stakes medical reasoning across Western and non-Western languages.
- Uses complementary benchmarks (medical-specific vs general) to probe domain-sensitive disparities.
- Identifies concrete failure categories (e.g., embryology, dosage conversion) relevant for targeted mitigation.

**Weaknesses:**

- Empirical confounds (inconsistent model versions, differing language sets, unstratified sampling) weaken causal claims about language vs domain effects.
- Lacks statistical testing and translation-based baselines to isolate multilingual comprehension from domain knowledge gaps.
- Presentation inconsistencies and limited structured error taxonomy reduce interpretability and reproducibility.

---

### Meta-Review · Area_Chair_mTig · 2026-02-26

**Recommendation:** Reject
**Confidence:** 4

**Metareview:**

This paper evaluates cross-linguistic disparities in medical reasoning using XMedBench and CrossMMLU, showing substantial variability across languages and highlighting fairness risks for multilingual clinical decision support. Reviewers find the topic timely and important, appreciate the use of complementary benchmarks, and value the effort to identify concrete failure categories. However, they raise several methodological issues that weaken the causal claims, including confounds from inconsistent model versions/language sets, unstratified sampling, and evaluating only the first 300 questions per language, as well as limited model coverage. Addressing these concerns would significantly strengthen the paper.

---

### Decision · Program_Chairs · 2026-03-02

Accept (Poster)